# Pollen Competition and Paternal Contribution during Artificially Controlled Pollination of Black Locust (*Robinia pseudoacacia* L.) without Castration

**Yuhan Sun** [1,†] **, Ruiyang Hu** [2,†] **, Li Dong** [1] **, Xiuyu Li** [1] **, Zijie Zhang** [1] **, Qi Guo** [1,3] **, Sen Cao** [1] **, Jiankang Li** [1] **, Peiyao Han** [1] **, Chao Han** [1] **, Saleem Uddin** [1] **, Cui Long** [1] **, Yingming Fan** [1] **and Yun Li** [1,*]

[1] Engineering Technology Research Center of Black Locust of National Forestry and Grassland Administration, National Engineering Laboratory for Tree Breeding, College of Biological Sciences and Technology, Beijing Forestry University, Beijing 100083, China; syh831008@163.com (Y.S.); dongli0408@126.com (L.D.); lixiuyu@bjfu.edu.cn (X.L.); zijiezhang@bjfu.edu.cn (Z.Z.); guoqi0529@126.com (Q.G.); caosen0419@bjfu.edu.cn (S.C.); jiankangli77@163.com (J.L.); hpy140912@126.com (P.H.); jiani.690@163.com (C.H.); saleemkhan86@hotmail.com (S.U.); longcui@bjfu.edu.cn (C.L.); yingming_1130@163.com (Y.F.)

[2] Experimental Center of Forestry in North China, Chinese Academy of Forestry, National Permanent Scientific Research Base for Warm Temperate Zone Forestry of Jiulong Mountain in Beijing, Beijing 102300, China; hury1102@163.com

[3] College of Agriculture, Henan University of Science and Technology, Luoyang 471023, China

[*] Correspondence: yunli@bjfu.edu.cn; Tel.: +86-10-6233-6094; Fax: +86-10-6233-6094

[†] These authors contributed equally to this article.

**Abstract:** (1) Background: Considering the serious damage caused by castration and the extremely high outcrossing rate in nature, we hypothesized that artificial controlled pollination of black locust without castration could be conducted for hybridization breeding. (2) Methods: The study conducted controlled pollination on 20 mating combinations of black locust without castration using a single or mixed male parent. Offspring of different developmental stages and the leaves of parents were collected to extract DNA and perform paternity analysis using SSR molecular markers. (3) Results: The contribution rate of each male parent differed according to developmental stage after pollination using different pollens mixed in equal proportions. There were significant correlations between the genetic similarity between each male parent and female parent and contribution rate of each male parent at three different developmental stages after pollination. (4) Conclusions: The composition of offspring pollen donors showed no bias toward selfing or outcrossing when artificially pollinated without castration. Hybrid breeding of black locust by artificially controlled pollination without castration may not be feasible, given that our manual method resulted in a large number of abortive and abnormal offspring. Introduction of honeybees in a limited space to conduct controlled pollination of black locust for hybrid breeding may be feasible.

**Keywords:** *Robinia pseudoacacia* L.; hybridization breeding; artificial controlled pollination; without castration; pollen competition; outcrossing rate; honeybee

## 1. Introduction

Black locust (*Robinia pseudoacacia* L.) is a fast-growing tree species native to the United States. It is adaptable and tolerates a wide range of soils including sandy or nearly barren soils [1,2]. It fixes nitrogen and produces good wood suitable for poles, beams, and firewood, as well as scented flowers for bees [1,2]. Compared with Populus [3,4], Eucommia [5], Eucalyptus [6], and other tree species, hybrid breeding of black locust is in its infancy. There has been no report of breeding of new varieties of black locust by artificial hybridization. However, the genetic variation among black locust individuals is greater than that among populations [7,8]. Hybrid breeding can make better use of genetic

variation among individuals for genetic improvement of black locust. The low seed-setting rate of black locust and the damage caused by castration hamper its hybridization [9,10]. The seed-setting rate becomes extremely low after castration of black locust, and therefore the offspring group cannot be selected. The seed-setting rate of black locust was only 3.56% when pollinated after castration [10]. In addition, as a typical insect-pollinated Papilionoideae species, the time-consuming and laborious castration, collection of pollen, and artificial pollination impede hybrid breeding of black locust. Castration damages the ovary of black locust. Artificially controlled pollination with castration is not only tedious but also has a very low seed-setting rate, which makes it impossible to obtain an adequate number of offspring populations for the selection of superior single plants. Therefore, other methods of hybrid breeding of black locust are needed. Surles et al. [11] reported that black locust is a highly outcrossing tree species. Moreover, Yuan [12] reported that the outcrossing rate of natural pollination is extremely high. The outcrossing rate of offspring seedlings can reach 95.83% when pollinated openly in nature. Under natural conditions, the offspring of black locust have an extremely high outcrossing rate, and therefore an adequate number of offspring populations can be obtained. Superior single plants in the offspring population can be selected and inbred individuals can be identified by paternity analysis based on simple sequence repeat (SSR) molecular markers to obtain hybrid offspring [11–13]. Because the outcrossing rate of offspring is high under natural conditions, can we ignore the effect of the female parent's pollen and conduct artificial hybridization of black locust without castration?

In general, plant mating systems promote outcrossing, increase genetic variation, and prevent inbreeding recession, and thus their evolution is an avoidance and conservation strategy for plants [14,15]. Although it is hypothesized that opposite evolutionary forces will lead to fully selfing or completely outcrossing systems, which are stable reproduction strategies [16,17], up to 42% of plant species have a mixed mating system, i.e., both self-crossing and outcrossing [18]. Mixed mating systems ensure reproductive success in unpredictable environments. In addition, the pollination mechanism of plants as a selective force regulates and determines whether pollen donors produce offspring [19,20]. In nature, not all pollen grains of this species that are transmitted to the stigma of the pistil are successfully inseminated and produce offspring [21]. There may be differences in the composition of offspring pollen donors at different developmental stages, which enhances the competitiveness of the population [21].

Considering the damage caused by castration and the extremely high outcrossing rate in nature, we hypothesized that artificial controlled pollination of black locust without castration could be conducted for hybridization breeding. Moreover, we questioned whether there are differences in the composition of offspring pollen donors at different developmental stages when pollinated without castration, and whether the genetic relationship between the male and female parents affects mating compatibility.

We conducted artificial controlled pollination on 20 mating combinations of black locust without castration using a single or mixed male parent. Offspring of different developmental stages and the leaves of parents were collected to extract DNA and perform paternity analysis using SSR molecular markers. We assessed the dynamic changes of the offspring pollen donors of black locust under controlled pollination without castration and evaluated the influence of the parental genetic relationship on the mating compatibility, as well as the feasibility of artificial hybridization of black locust without castration. Our results will facilitate artificial hybridization of black locust without castration.

## 2. Materials and Methods

### 2.1. Controlled Pollination and DNA Extraction

From 2016 to 2018, black locusts in Mijiabao nursery (Yanqing, Beijing, China) were selected as the female and male parents of the hybrid combinations in May. Before flowering, maternal flowers were bagged (Figure 1) without castration to protect from foreign pollen. Paternal flowers were picked to collect pollen manually during pollen dispersal. Single or

mixed paternal pollen was used for artificial controlled pollination during the receptive period of the maternal stigmas. Pollen vitality was assayed by the triphenyl tetrazolium chloride staining method before pollination. The mating combinations are shown in Table 1.

In 2016 and 2017, immature zygotic embryos of each cross combination at 7 and 30 days after pollination were collected and stored in an ultra-low-temperature freezer (−80 °C). A Leica M205A stereoscope was used for morphologic observation. DNA was extracted from immature zygotic embryos using a Micro/Clinical Genomic DNA Rapid Extraction Kit (Aidlab Biotechnologies Co., Ltd., Beijing, China). In 2018, mature seeds of each combination were collected. A DNAsecure Plant Kit (Tiangen Biotech (Beijing) Co., Ltd., Beijing, China) was used to extract the DNA of the mature seeds and the leaves of the parents. The number of samples for each combination at different development stages ranged from 80 to 330.

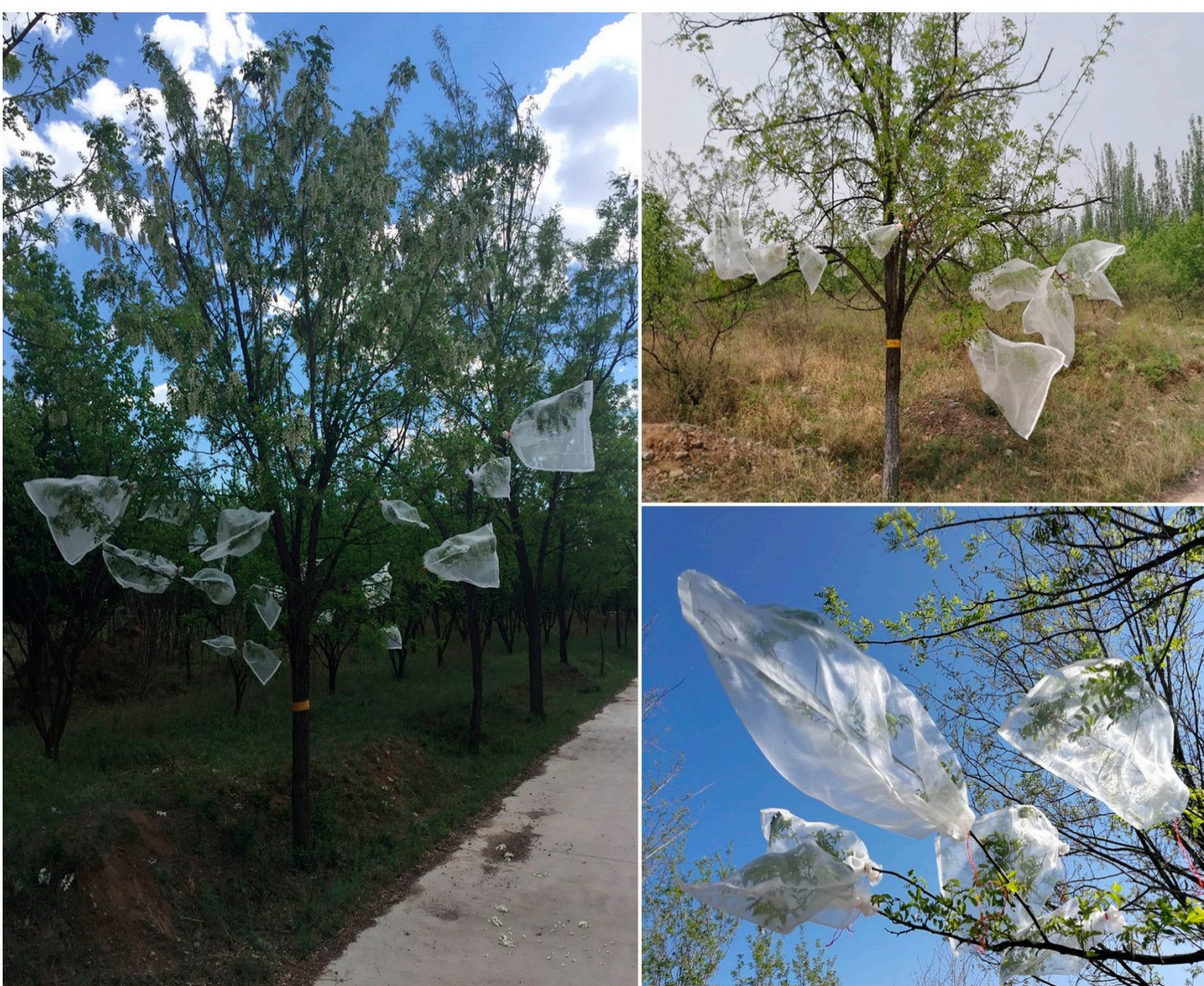

**Figure 1.** Maternal flowers were bagged without castration before flowering.

**Table 1.** Mating combinations of controlled pollination in each year.

| Year | Female Parents (♀) | Male Parents (♂) | Mating Combinations |
|---|---|---|---|
| 2016 | 2016-C1<br>2016-C2<br>2016-C3 | 2016-C1 + 2016-C4 + 2016-C5<br>2016-C2 + 2016-C4 + 2016-C5<br>2016-C3 + 2016-C4 + 2016-C5 | 2016-C1 × (2016-C1 + 2016-C4 + 2016-C5)<br>2016-C2 × (2016-C2 + 2016-C4 + 2016-C5)<br>2016-C3 × (2016-C3 + 2016-C4 + 2016-C5) |
| 2017 | 2017-C1<br>2017-C2<br>2017-C3 | 2017-♂1 + 2017-♂2 + 2017-♂3 | 2017-C1 × (2017-♂1 + 2017-♂2 + 2017-♂3)<br>2017-C2 × (2017-♂1 + 2017-♂2 + 2017-♂3)<br>2017-C3 × (2017-♂1 + 2017-♂2 + 2017-♂3) |
| | 2017-C4<br>2017-C5<br>2017-C6 | 2017-♂4 + 2017-♂5 + 2017-♂6 | 2017-C4 × (2017-♂4 + 2017-♂5 + 2017-♂6)<br>2017-C5 × (2017-♂4 + 2017-♂5 + 2017-♂6)<br>2017-C6 × (2017-♂4 + 2017-♂5 + 2017-♂6) |
| | 2017-C7<br>2017-C8<br>2017-C9 | 2017-♂7 + 2017-♂8+2017-♂9 | 2017-C7 × (2017-♂7 + 2017-♂8 + 2017-♂9)<br>2017-C8 × (2017-♂7 + 2017-♂8 + 2017-♂9)<br>2017-C9 × (2017-♂7 + 2017-♂8 + 2017-♂9) |
| 2018 | 2018-C1<br>2018-C4<br>2018-C5 | 2018-♂4 | 2018-C1 × 2018-♂4<br>2018-C4 × 2018-♂4<br>2018-C5 × 2018-♂4 |
| | 2018-C2<br>2018-C3 | 2018-♂2 | 2018-C2 × 2018-♂2<br>2018-C3 × 2018-♂2 |
| | 2018-C6<br>2018-C7<br>2018-C8 | 2018-♂1 + 2018-♂3 + 2018-♂5 | 2018-C6 × (2018-♂1 + 2018-♂3 + 2018-♂5)<br>2018-C7 × (2018-♂1 + 2018-♂3 + 2018-♂5)<br>2018-C8 × (2018-♂1 + 2018-♂3 + 2018-♂5) |

*2.2. PCR Amplification with SSR Primers*

Four expressed sequence tag (EST)-SSR and four genomic SSR markers [22,23] (Table 2) were selected for PCR amplification using an ABI9700 Thermal Cycler (Applied Biosystems, Carlsbad, CA, USA) in a 20 μL reaction mixture containing 1 μL of genomic DNA (25 ng/μL), 10 μL of 2× TSINGKE® Master Mix (blue) (Beijing TsingKe Biotech Co., Ltd., Beijing, China), 7 μL of double-distilled $H_2O$, and 0.5 μmol/L of each primer. The forward primer was labeled with a fluorescent dye (FAM, HEX, or ROX; Beijing Rui Biotech Co., Ltd., Beijing, China) during synthesis. Amplification was performed as follows: initial denaturation at 94 °C for 5 min, followed by 10 cycles of 94 °C for 30 s, 63–53 °C for 30 s (decreasing by 1 °C per cycle), and 72 °C for 90 s; 20 cycles at 94 °C for 30 s, 58 °C for 30 s, and 72 °C for 90 s; and a final extension for 10 min at 72 °C [24]. The PCR products were separated by capillary electrophoresis using an ABI3100 DNA analyzer (Applied Biosystems), and amplicon fragments were sized using GeneMarker 1.8.0 software (SoftGenetics, State College, PA, USA).

*2.3. Paternity and Data Analysis*

The capillary electrophoresis results were applied to paternity analysis with Cervus 3.0 software to identify the most likely male parent [13,25]. After allele frequency analysis and a simulated paternity analysis, paternity analysis was performed using at least six of eight pairs of SSR primers matching the most likely male parent. The parameters of paternity analysis were as follows: number of cycles, 10,000; proportion of candidate fathers sampled, 1.000; proportion of mistyped loci, 0.010; confidence level, 80–95% [12]. NTSYS pc2.1 software (Applied Biostatistics Inc, New York, NY, USA) was used to calculate Nei's genetic similarity with the SM (simple matching) coefficient [26]. Microsoft Excel 2016 (Microsoft Corp., Redmond, WA, USA) and SPSS 24.0 (IBM Corp., Armonk, NY, USA) were used for statistical analysis.

**Table 2.** Information of four EST-SSR and four genomic-SSR markers.

| Locus | Repeat Motif | Primer Sequences (5′ to 3′) | Size (bp) | Source |
|---|---|---|---|---|
| Rply3 | (GTGGTA)$_4$ | F: GCCTCATAAATAAAAGGAACG | 138–164 | Our group |
| | | R: CTGCCATTGGTAACTGGTAAA | | |
| Rply5 | (ATG)$_8$ | F: GAGTCATGCCCTTTGTATGTT | 122–143 | Our group |
| | | R: TGTCACCTTCAAGTCCCTATT | | |
| Rply8 | (CT)$_{14}$ | F: TCCCTACATAAAACTCCAAA | 239–271 | Our group |
| | | R: TCATTAAGTCAGCACTCACAG | | |
| Rply49 | (CT)$_8$ | F: CCCCGTACAGTTCCATCT | 118–144 | Our group |
| | | R: GACCTCGTAAAAGCCACC | | |
| Rops08 | (CA)$_8$TA(CA)$_3$ | F: TTCTGAGGAAGGGTTCCGTGG | 191–205 | Lian et al., 2002 |
| | | R: GTTAAAGCAACAGGCACATGG | | |
| Rp109 | (AG)$_{17}$ | F: GAGGAATCACAAAACCGTTTGG | 136–160 | Mishima et al., 2009 |
| | | R: TGGGATTTGAGAGAGTGGTGGTG | | |
| Rp200 | (AG)$_{23}$ | F: GGTTTCTTTGTTCACCTGCTCTGG | 160–198 | Mishima et al., 2009 |
| | | R: ACCTACGTGTCCACGGCTCT | | |
| Rp206 | (GT)$_9$ | F: GCCAAATCCCATTAGAT-CACAGTTGA | 222–246 | Mishima et al., 2009 |
| | | R: AGAAGTTAGACTTACGTGCTGC | | |

## 3. Results

### 3.1. Early Development of Offspring

In 2016 and 2017, immature zygotic embryos of three and nine cross combinations at 7 and 30 days after pollination were collected, respectively. In 2016, the average maximum temperature for 7 days after pollination was 24.43 °C, and the average minimum temperature was 10.14 °C. For 2017, it was 29.43 and 15.14 °C, respectively. The developmental process of immature zygotic embryos varied greatly according to the average highest and lowest temperature in the field environment within 7 days after pollination. At 7 and 30 days after pollination, the ovary and immature zygotic embryos in 2017 were significantly larger than those in 2016 (Figures 2 and 3). Moreover, the immature zygotic embryos in 2017 began to be aborted at 7 days after pollination (Figure 2b,e). Abortion occurred in the immature zygotic embryos of all cross combinations at 30 days after pollination (Figure 2c,d). These results indicate that the ambient temperature after pollination affects the development of black locust seeds, and high temperatures accelerate their development. Offspring selection began early in the development of black locust seeds. Some offspring were eliminated shortly after pollination due to abortion.

### 3.2. Composition of Immature Zygotic Embryo Pollen Donors According to Developmental Stage

The paternity analysis of the immature zygotic embryos at different developmental stages of the 12 cross combinations in 2016 and 2017 showed no obvious trends in the composition of their pollen donors. The proportions of the offspring derived from outcross pollen of each cross combination at 7 days after pollination were 57.19%, 98.68%, 66.56%, 12.58%, 95.97%, 79.32%, 59.37%, 84.38%, 44.80%, 97.89%, 76.04%, and 58.33%. The outcrossing rate of 10 of the 12 cross combinations exceeded 50% (Figure 4). At 30 days after pollination, the proportions were 100.00%, 13.75%, 10.45%, 0.73%, 11.14%, 52.86%, 13.54%, 23.96%, 15.63%, 60.42%, 85.42%, and 63.54%, respectively. The outcrossing rate of 5 of the 12 cross combinations was greater than 50% (Figure 5). At two different developmental stages, the outcrossing rate of nine combinations decreased at 30 days after pollination compared to at 7 days.

### 3.3. Development and Composition of Pollen Donors of Aborted Seeds

In 2018, we conducted artificially controlled pollination without castration on eight cross combinations, expecting to obtain mature seeds to grow into seedlings. Thus, we extracted DNA and performed a paternity analysis using the leaves of the seedlings. Unfortunately, almost all ripe seeds of each combination were aborted or deformed. Because of the large number of aborted seeds produced by self-crossing of black locust in the open pollination state [12], we suspected that the large number of aborted and deformed seeds following artificially controlled pollination was caused by the high selfing rate of the offspring. Therefore, we extracted DNA and performed a paternity analysis on these aborted and deformed seeds. The results showed that the outcrossing rates of the eight combinations were 71.58%, 82.61%, 45.83%, 94.79%, 51.04%, 75.79%, 39.58%, and 33.33% (Figure 6). The reason for the large number of aborted and deformed seeds was not the high selfing rate. In the combination with an outcrossing rate of 94.79%, the offspring had a large number of abnormal and abortive seeds at maturity.

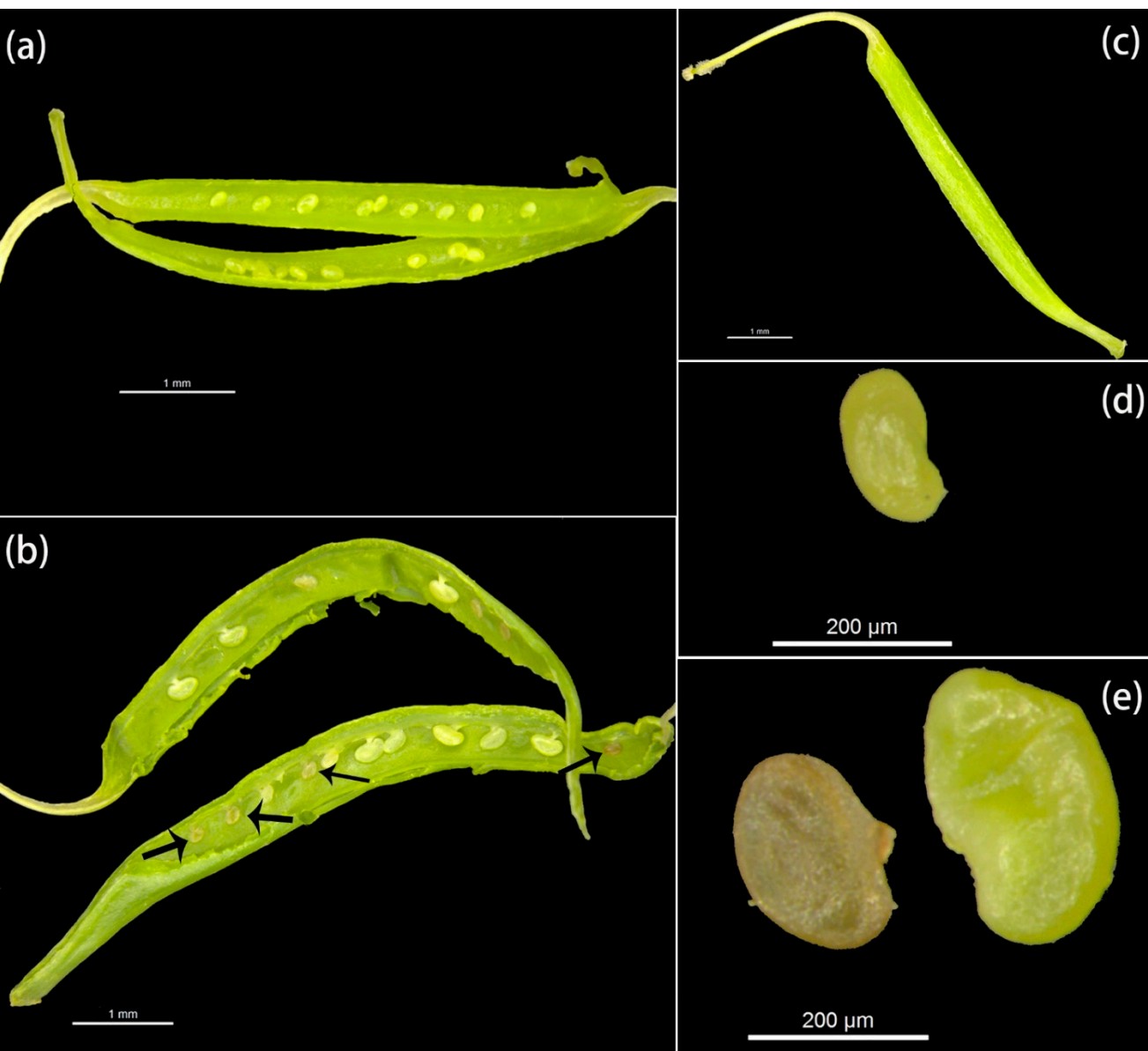

**Figure 2.** Ovaries and immature zygotic embryos at 7 days after pollination in (**a**,**c**,**d**) 2016 and (**b**,**e**) 2017.

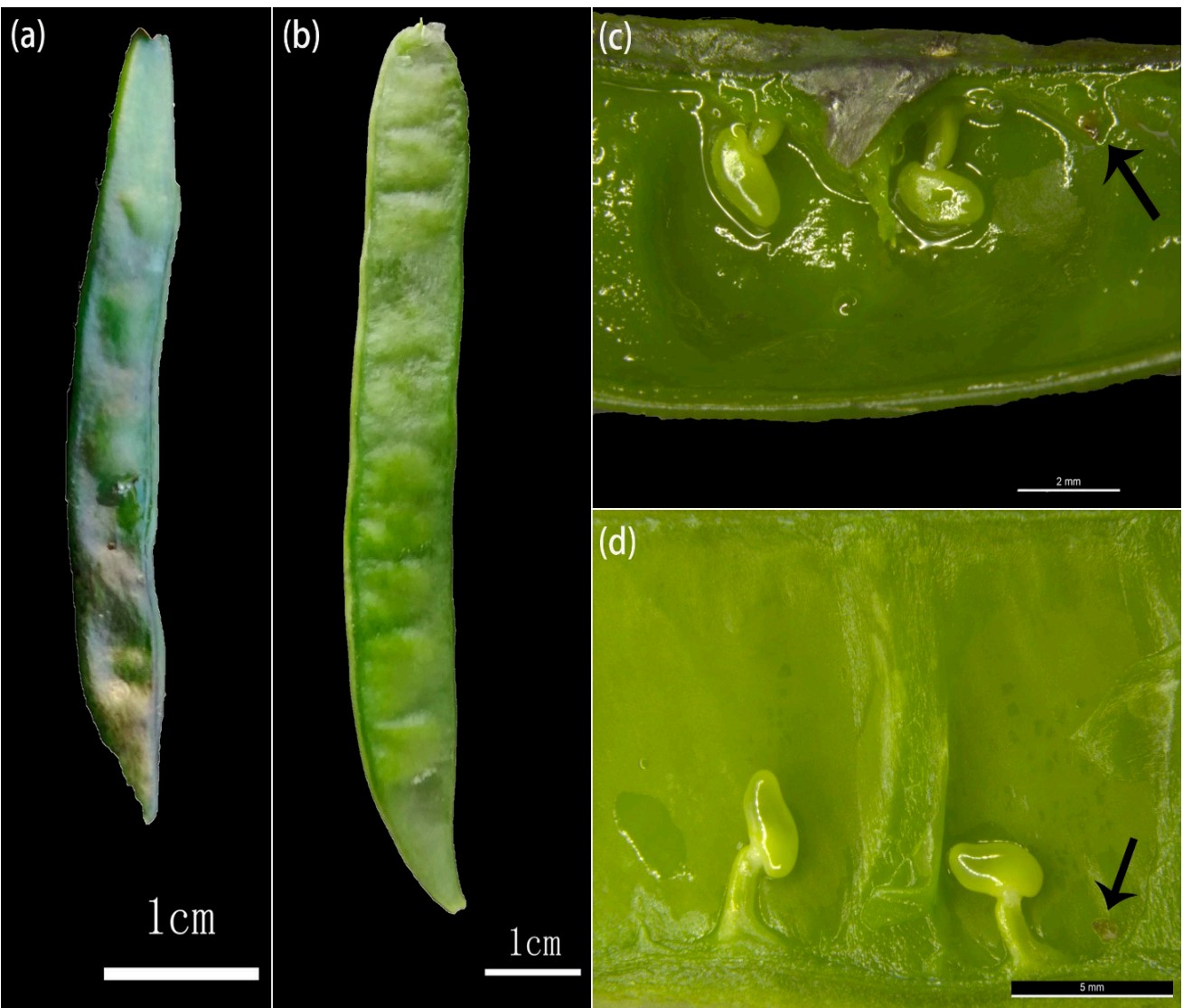

**Figure 3.** Ovaries and immature zygotic embryos at 30 days after pollination in (**a**,**c**) 2016 and (**b**,**d**) 2017.

### 3.4. Contribution Rate and Relationship between Genetic Similarity and Hybrid Affinity of Parents

We conducted a statistical analysis of the contribution rate of each male parent at different stages after pollination of each hybrid combination and calculated the genetic similarity between each male parent and the female parent (Figures 7–11). The contribution rate of each male parent was different at three developmental stages after pollination using different pollens mixed in equal proportions. For example, the mixed male parents for the combination ♀2016-C3 were 2016-C4 and 2016-C5. The contribution rates for these two male parents at 7 days after pollination were 23.61% and 2.62%, respectively (Figure 7e). At 30 days after pollination, the contribution rates were 7.47% and 1.49%, respectively (Figure 7f). A small amount of pollen from other female parents infected during artificial pollination may have a high fertilization rate in many combinations. The contribution rate of 2016-C2 reached 38.03% at 7 days after pollination in the combination ♀2016-C3 (Figure 7e). The difference between the contribution rate of each male parent at different stages after pollination of each hybrid combination indicated that pollen competition and offspring selection occurred throughout the development of seed of black locust. In addition, the amount of pollen from different male parents during pollination was not related to the fertilization rate.

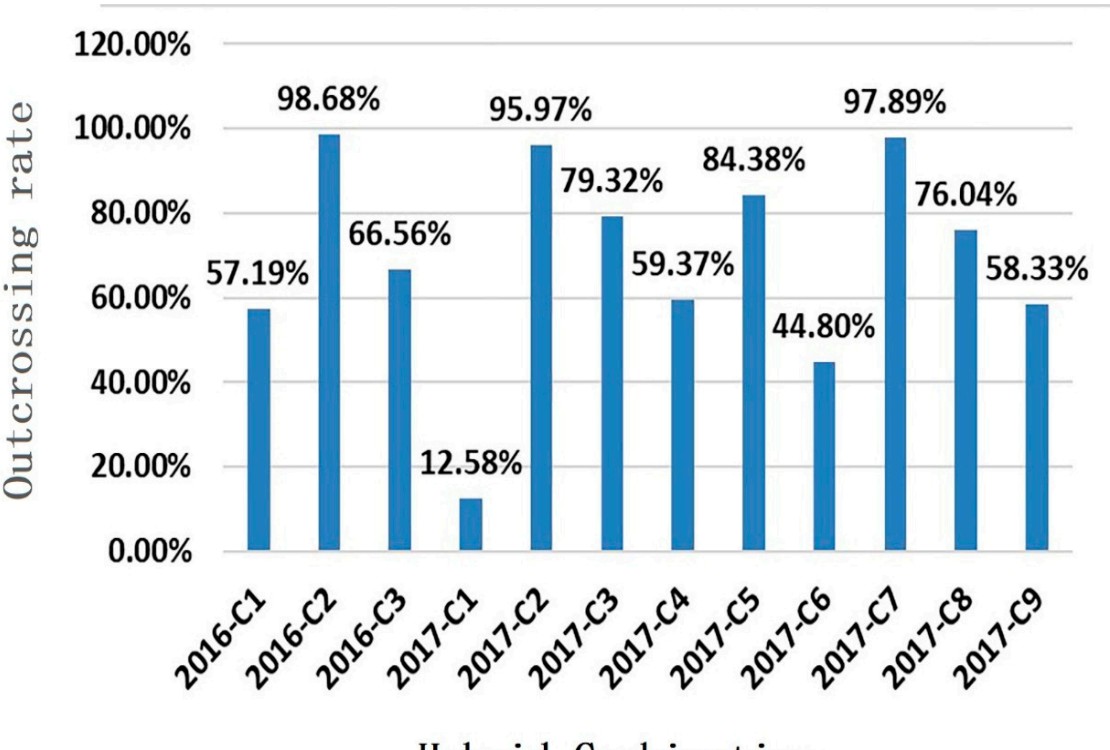

**Figure 4.** Outcrossing rate of each combination at 7 days after pollination in 2016 and 2017.

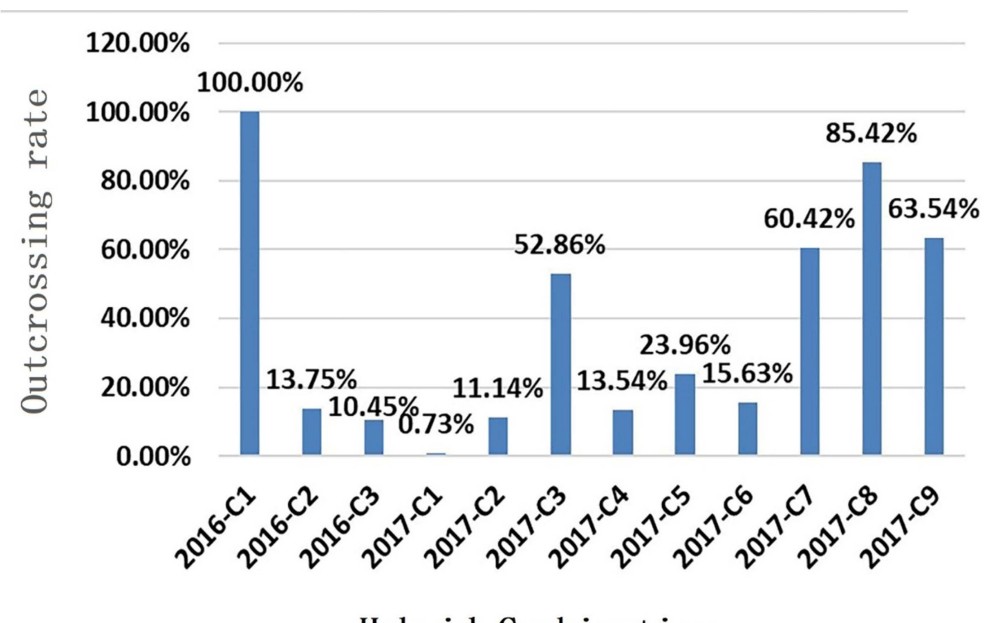

**Figure 5.** Outcrossing rate of each combination at 30 days after pollination in 2016 and 2017.

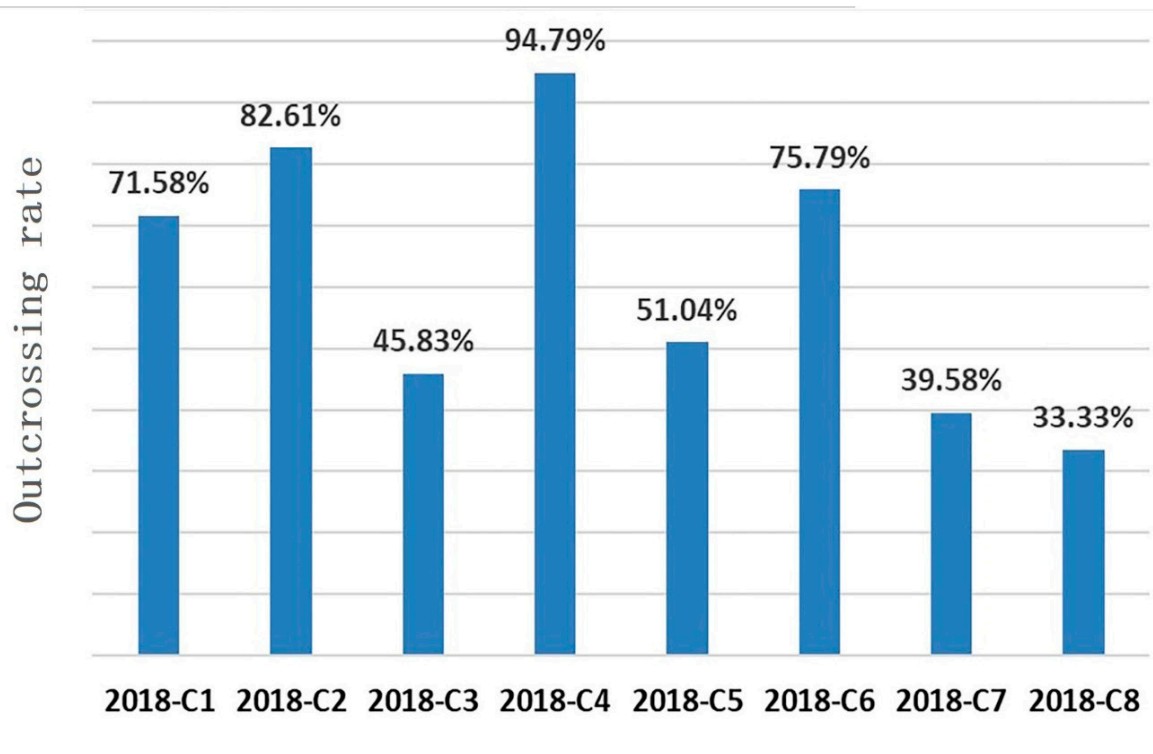

**Figure 6.** Outcrossing rate for aborted and deformed seeds of each combination in 2018.

There were significant correlations between the genetic similarity between each male parent and female parent and contribution rate of each male parent at three different developmental stages after pollination (Table 3). As the contribution rate of each male parent can reflect the mating compatibility, this suggests that the genetic relationship between the male parent and the female parent affects the mating compatibility. However, the relationship between them was not linear.

**Table 3.** Correlation of the genetic similarity and mating affinity of the male and female parents at different stages after pollination.

| Stages | Number of Samples | Correlation | Genetic Similarity | Hybrid Affinity |
|---|---|---|---|---|
| 7 days after pollination | 69 | Pearson correlation | 1 | 0.272 * |
| | | Significance (two-tailed test) | | 0.024 |
| 30 days after pollination | 69 | Pearson correlation | 1 | 0.676 ** |
| | | Significance (two-tailed test) | | 0.000 |
| Seed maturity | 36 | Pearson correlation | 1 | 0.446 ** |
| | | Significance (two-tailed test) | | 0.006 |

Note: *, $p < 0.05$; **, $p < 0.01$.

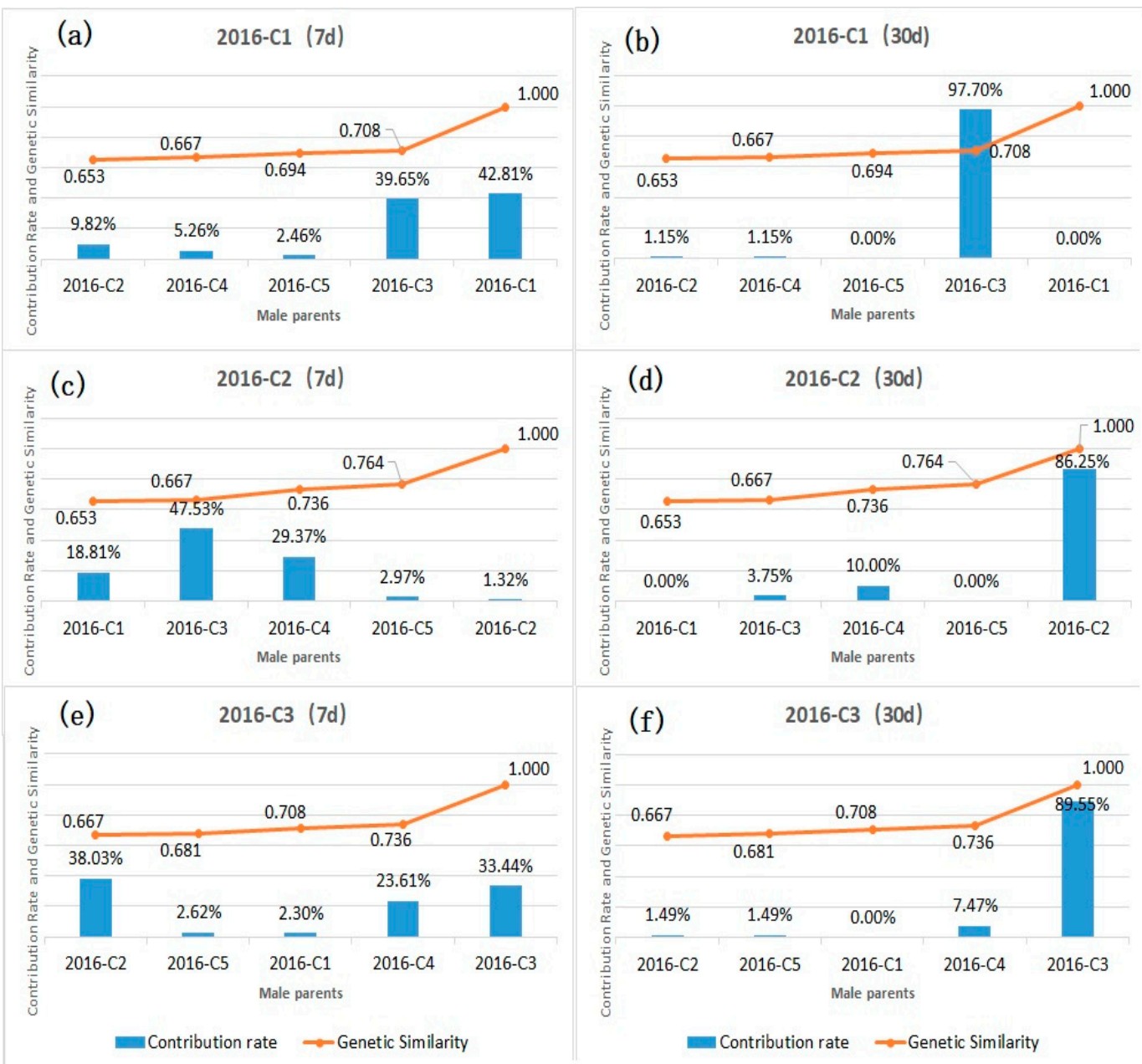

**Figure 7.** Contribution rate of each male parent and genetic similarity between each male parent and the female parent for the hybrid combinations: (**a**) ♀2016-C1, (**c**) ♀2016-C2, and (**e**) ♀2016-C3 at 7 days after pollination, and (**b**) ♀2016-C1, (**d**) ♀2016-C2, and (**f**) ♀2016-C3 at 30 days after pollination.

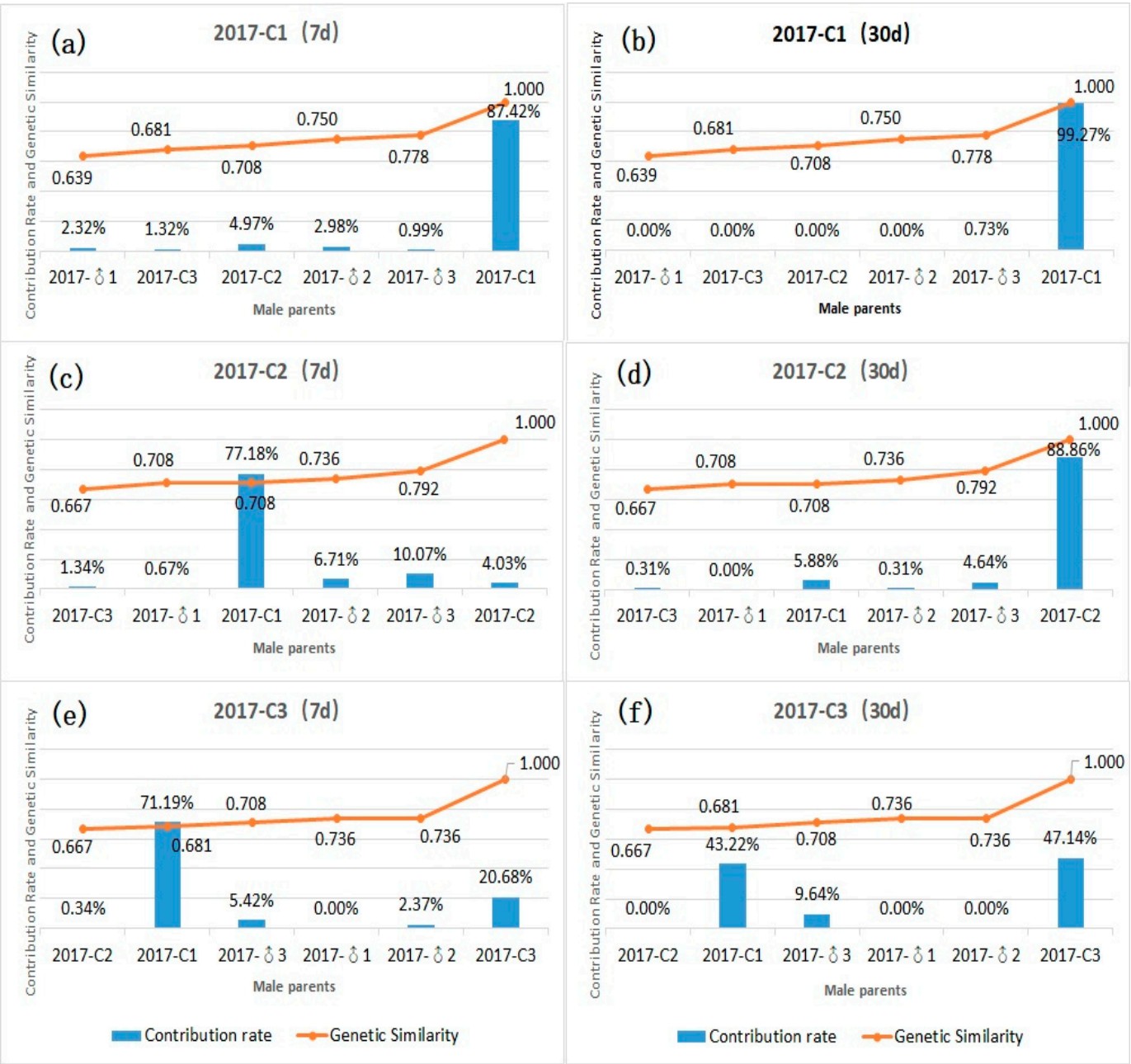

**Figure 8.** Contribution rate of each male parent and genetic similarity between each male parent and the female parent for the hybrid combinations: (**a**) ♀2017-C1, (**c**) ♀2017-C2, and (**e**) ♀2017-C3 at 7 days after pollination, and (**b**) ♀2017-C1, (**d**) ♀2017-C2, and (**f**) ♀2017-C3 at 30 days after pollination.

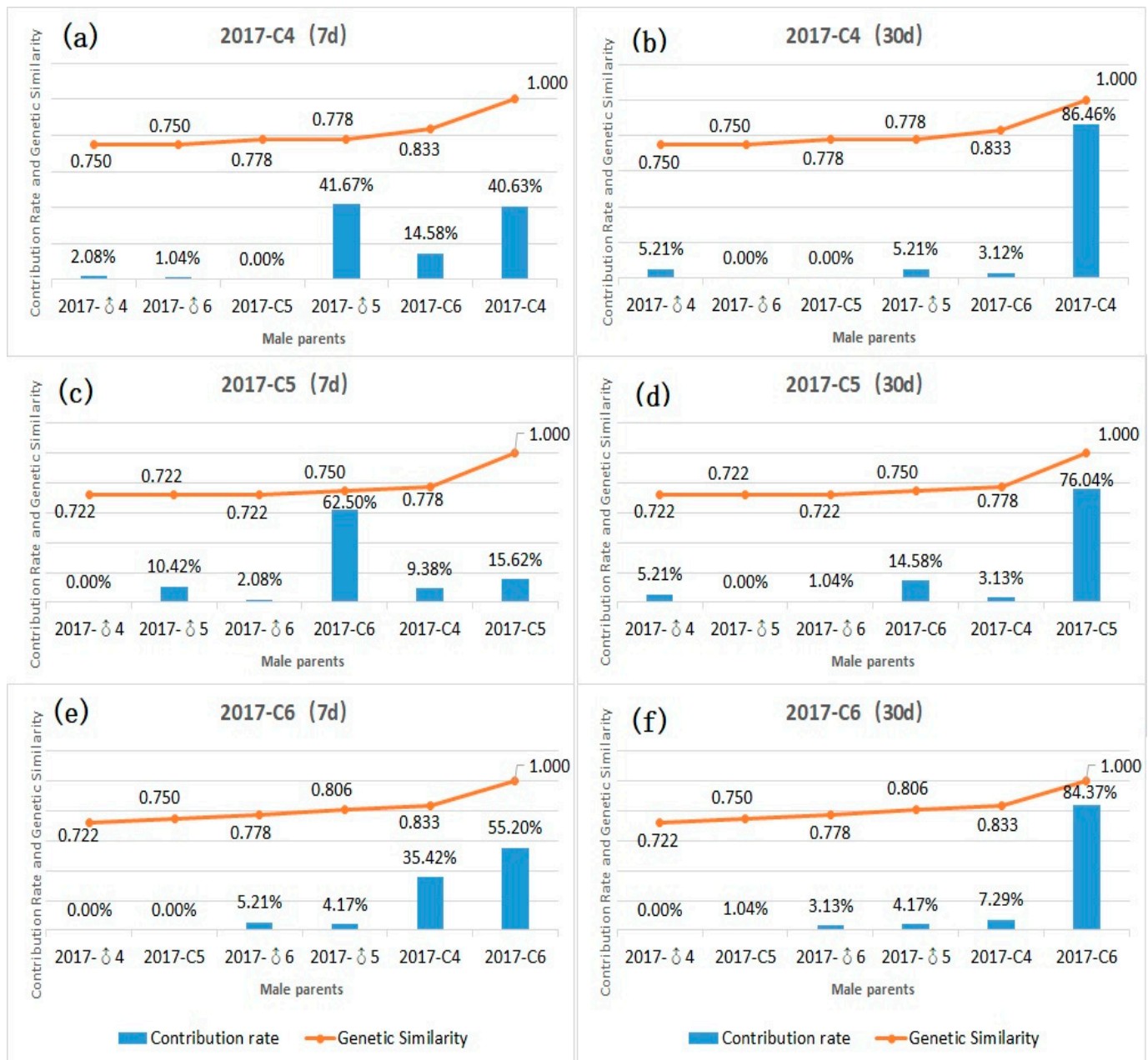

**Figure 9.** Contribution rate of each male parent and genetic similarity between each male parent and the female parent for the hybrid combinations: (**a**) ♀2017-C4, (**c**) ♀2017-C5, and (**e**) ♀2017-C6 at 7 days after pollination, and (**b**) ♀2017-C4, (**d**) ♀2017-C5, and (**f**) ♀2017-C6 at 30 days after pollination.

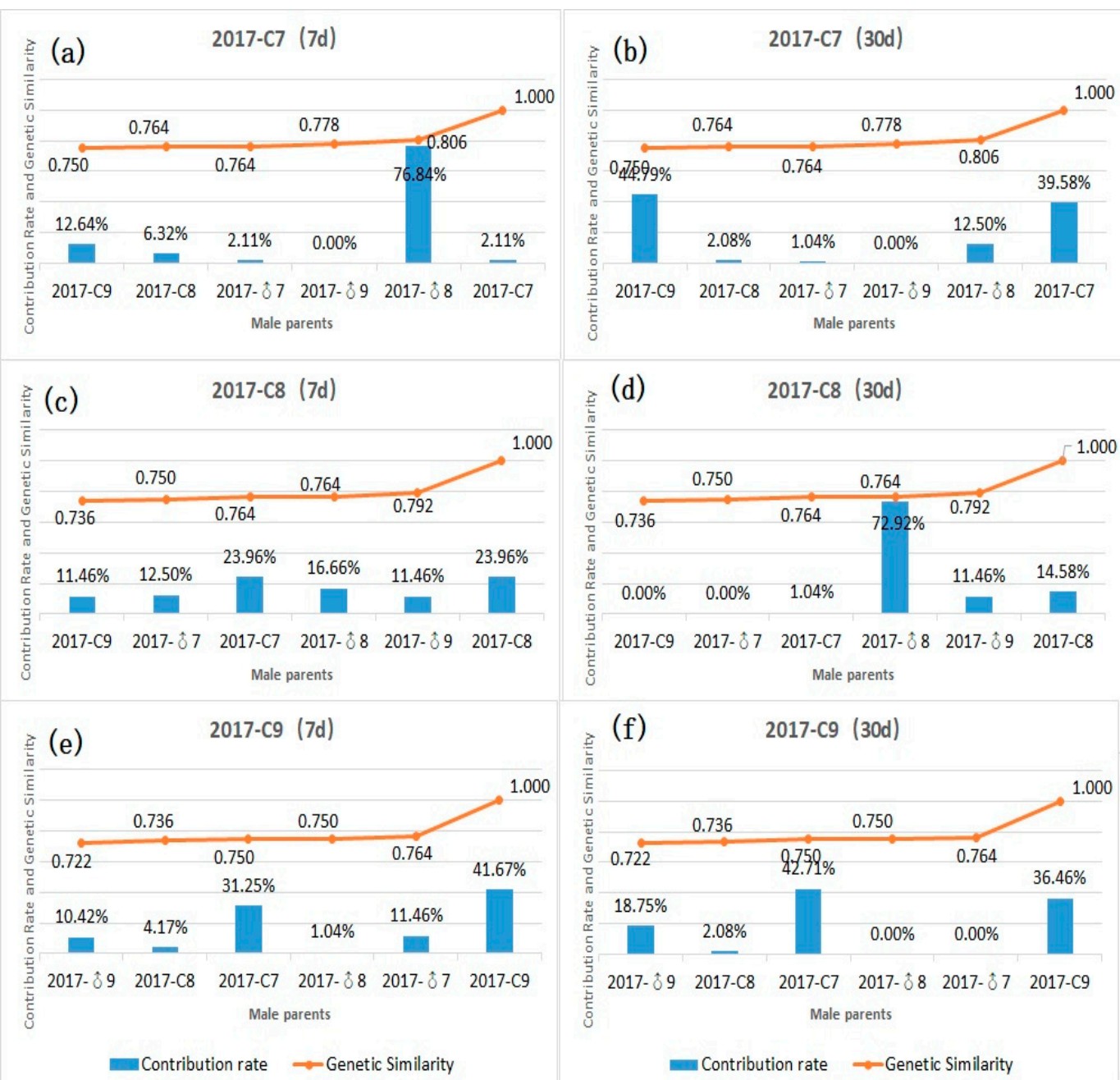

**Figure 10.** Contribution rate of each male parent and genetic similarity between each male parent and the female parent for the hybrid combinations: (**a**) ♀2017-C7, (**c**) ♀2017-C8, and (**e**) ♀2017-C9 at 7 days after pollination, and (**b**) ♀2017-C7, (**d**) ♀2017-C8, and (**f**) ♀2017-C9 at 30 days after pollination.

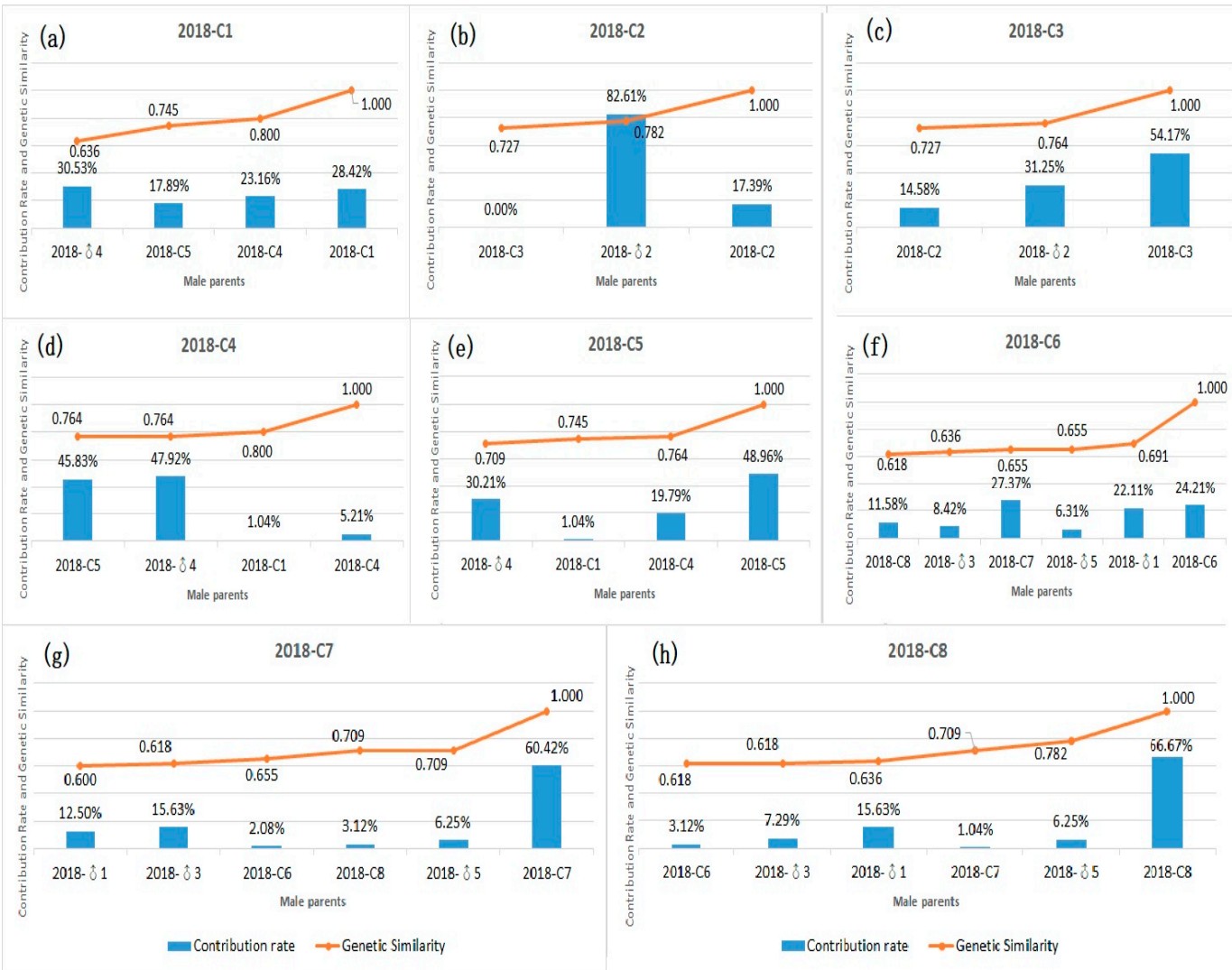

**Figure 11.** Contribution rate of each male parent and genetic similarity between each male parent and the female parent for the hybrid combinations: (**a**) ♀2018-C1, (**b**) ♀2018-C2, (**c**) ♀2018-C3, (**d**) ♀2018-C4, (**e**) ♀2018-C5, (**f**) ♀2018-C6, (**g**) ♀2018-C7, and (**h**) ♀2018-C8.

## 4. Discussion

### 4.1. Pollen Competition and Offspring Selection

Competition between pollens occurs when the amount of pollen on a plant stigma is greater than the number of ovules that need to be fertilized [27,28]. Competition between single pollen grains selects against harmful mutations because a large number of micro-gametophytic genes are expressed during pollen-tube growth [28–30]. The resulting pollen competition results in elongated pistils, enlarged stigmatic surfaces, and delayed stigma receptivity and functional syncarpy to produce higher-quality offspring [28,29,31–37]. This will lead to differences in pollen donor composition at different developmental stages of the offspring.

Yuan analyzed the outcrossing rate of aborted seeds, mature seeds, and offspring seedlings obtained under open pollination conditions of black locust and predicted pollen competition and offspring selection during development [12]. In this study, we observed immature zygotic embryos of black locust at 7 and 30 days after pollination and conducted a paternal analysis. The results confirmed the occurrence of pollen competition and offspring selection during the development of offspring of black locust. DNA extraction from the immature zygotic embryos was hampered by their small size and the large number of

samples. After isolating individual immature zygotic embryos under a stereomicroscope, we extracted their DNA using a kit not commonly used for plants.

Pollen competition and offspring selection began during early development after controlled pollination without castration and occurred throughout the development of the offspring. The pollen donor composition of the offspring of the same mating combination at different developmental stages after pollination differ markedly due to adoption of a mixed paternal rather than a single paternal parent. The impact of pollen competition is greater when competition occurs among several pollen donors than among pollen from a single donor [28,38].

### 4.2. Mating System of Black Locust

Dini-Papanastasi et al. [39] and Surles et al. [11] reported that black locust is a highly outcrossing tree species, and no seeds can be obtained by selfing. However, Sun [10] and Yuan [12] showed that black locust has a mating system based on outcrossing, but selfing can also produce offspring. In this study, the outcrossing rate of 10 of 12 cross combinations exceeded 50% at 7 days after pollination, and that of 5 of 12 cross combinations exceeded 50% at 30 days after pollination. For abnormal and abortive seeds at maturity, the outcrossing rate of five of eight cross combinations exceeded 50%. These results suggest that black locust has a mixed mating system—both self-crossing and outcrossing. However, external factors (e.g., manual methods) may affect the proportion of selfing and outcrossing offspring. This may explain the discrepancy between our results and prior works. It is reported that up to 42% of plant species have a mixed mating system to ensure reproductive success in unpredictable environments [18].

### 4.3. Controlled Pollination of Black Locust without Castration

Due to its low seed-setting rate and the damage caused by castration, hybridization of black locust is problematic [9–12,39]. Therefore, we conducted controlled pollination without castration. Among the 20 cross combinations, whether at 7 days after pollination, 30 days after pollination, or at the mature seed stage, although the proportion of selfing and outcrossing in the offspring changed to some extent, there was no bias toward selfing or outcrossing. This indicates that black locust produces hybrids following controlled pollination without castration. However, almost all seeds were abnormal or abortive, possibly due to manual methods. In nature, black locust is dependent on pollination by Hymenoptera insects [10–12], and therefore artificially controlled pollination may not be suitable and artificially controlled pollination may be insufficient. Hybrid breeding of black locust by artificially controlled pollination may not be feasible. Insect pollination is an important part of natural ecosystems and plays a major role in agroecosystems [40]. Animal pollinators increase seed or fruit production by approximately 85% of crops worldwide [40]. The low fruit-setting rate of black locust may be due to an insufficient number of pollinating insects.

Pollination by wild insects is better than that by honeybee, but the honeybee makes the greatest contribution to crop pollination [40]. Wild pollinating insects are affected by external factors, such as farming methods, habitat destruction, pests, and climate change. Artificially grown honeybees are less affected by these factors, and their application to crop pollination yields relatively stable outcomes. This is the reason why the demand for honeybee pollination by crops has increased annually [41,42].

Honeybee has been used to pollinate crops to improve yield and fruit quality. Chu et al. [43] reported that the sterility rate of Helianthus annuus L. pollinated by honeybee was 71.9% lower than that of the honeybee-free isolation zone, and the yield was increased 14-fold. Ge et al. [44] evaluated the pollination of soybeans (Glycine max (Linn.) Merr.) in production fields by introducing honeybee. The soybean yield in honeybee areas increased by 18.7–20.1%, and the 1000-kernel weight of soybeans increased by 6.0%. Shi et al. [45] conducted a honeybee pollination experiment on Brassica napus L.; the yield in the honeybee pollination area was 40.16% higher than that in the natural pollination area, and 114.98% higher than that in the honeybee-free pollination area.

For about six years, our research team has made attempts in this field. Before flowering of black locust, we used a large insect net, into which honeybees were introduced, to cover the trees. We harvested seeds and successfully cultivated seedlings in 2018 (Figure 12).

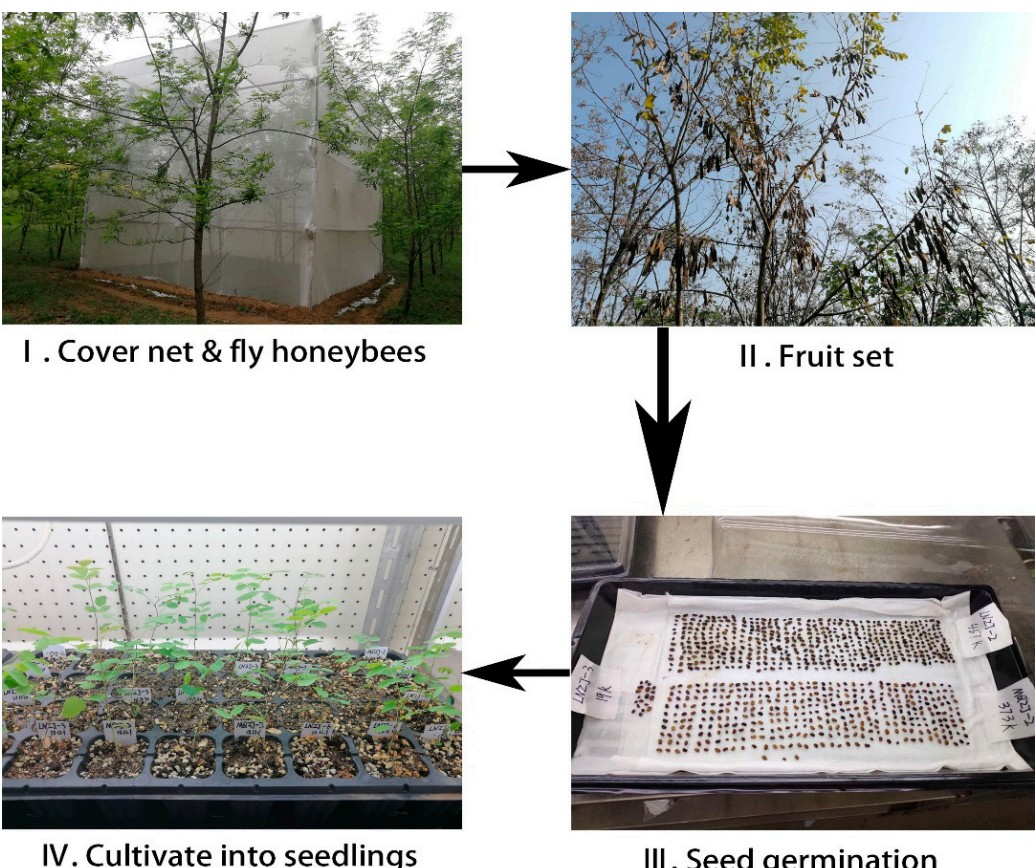

**Figure 12.** Process of introduction of honeybees to conduct controlled pollination.

## 5. Conclusions

Pollen competition and offspring selection begins early and occurs throughout the development of black locust seeds. The genetic relationship between the male and female parents of black locust affects the mating compatibility. However, the amount of pollen from the male parent during pollination was not found to be related to the fertilization rate. The composition of offspring pollen donors showed no bias toward selfing or outcrossing when artificially pollinated without castration. Hybrid breeding of black locust by artificially controlled pollination without castration may not be feasible, given that our manual method resulted in a large number of abortive and abnormal offspring. Introduction of honeybees in a limited space to conduct controlled pollination of black locust for hybrid breeding may be feasible.

**Author Contributions:** Conceptualization, Y.S. and Y.L.; methodology, Y.L.; software, L.D. and Z.Z.; investigation, X.L., Q.G., Y.S., S.C. and J.L.; data curation, P.H., C.H., S.U., Y.F. and C.L.; writing—original draft preparation, Y.S. and R.H.; writing—review and editing, Y.S. and R.H.; supervision, Y.L.; project administration, Y.L. All authors have read and agreed to the published version of the manuscript.

**Funding:** This research was funded by the Fundamental Research Funds for the Central Universities (2015ZCQ-SW-03), National Key R&D Program of China (2017YFD0600503), and National Nature Science Foundation of China (31570677).

**Institutional Review Board Statement:** Not applicable.

**Informed Consent Statement:** Not applicable.

**Conflicts of Interest:** The authors declare no conflict of interest. The funders had no role in the design of the study; in the collection, analyses, or interpretation of data; in the writing of the manuscript; or in the decision to publish the results.

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
