# Peer review of "Pollen Competition and Paternal Contribution during Artificially Controlled Pollination of Black Locust (Robinia pseudoacacia L.) without Castration"

_forests, doi:10.3390/f12101416_

Round 1
Reviewer 1 Report
Presented paper describes an attempt to perform controlled pollination in Robinia pseudacacia without castration, which causes widespread damage. The paper is clear and scientifically sound, however I would like to adress some issues:
- Authors chose the Nei's genetic similarity with simple matching coefficient. I am not sure, if this is the best method for measuring similarity using SSR markers. The reason for choosing this coefficient needs to be clarified.
- The designations of parental plants and mating combinations are hard to follow, because some plants and mating combinations share the same symbols. This makes figures 7-11 also less clear.
- Tables 3-5 could be joined into single table and the asterisk codes should be explained at the table footer. It is also not clear, if genetic similarity presented here is a mean value or, the highest measured.
- In figures 7-11, seven decimal places for genetic similarity are not necessary. Values should be rounded to three places.
Author Response
Response to Reviewer 1 Comments
Point 1: Authors chose the Nei's genetic similarity with simple matching coefficient. I am not sure, if this is the best method for measuring similarity using SSR markers. The reason for choosing this coefficient needs to be clarified.
Response 1: At present, Nei’s genetic similarity coefficient is widely used by researchers to measure the degree of genetic similarity between different individuals in the same population, which is used as the basis for judging the genetic relationship between different individuals. The individual genotype data obtained through SSR molecular marker amplification is widely used to calculate Nei’s genetic similarity coefficient. Therefore, in this study, Nei’s genetic similarity coefficient was used to analyze the genetic similarity between different varieties of Robinia pseudoacacia to judge the genetic relationship between the male parent and female parent of each Robinia pseudoacacia.
Point 2: The designations of parental plants and mating combinations are hard to follow, because some plants and mating combinations share the same symbols. This makes figures 7-11 also less clear.
Response 2: In Figure 7-11, the labeling of parents and hybrid combinations has been modified to make it clearer.
Point 3: Tables 3-5 could be joined into single table and the asterisk codes should be explained at the table footer. It is also not clear, if genetic similarity presented here is a mean value or, the highest measured.
Response 3: Table 3, table 4 and table 5 have been merged into one table and marked with asterisks. The genetic similarity is a mean value.
Point 4: In figures 7-11, seven decimal places for genetic similarity are not necessary. Values should be rounded to three places.
Response 4: The genetic similarity values in Figure 3-7 have been rounded, and the values have been reserved to only three decimal places.
Reviewer 2 Report
Authors report the differences in the composition of offspring pollen donors at various developmental stages when pollinated without castration, and also the compatibility effect of genetic relationship between male and female parents in black locust hybridization.
- The background and purposes are clear.
- The method and design are significant.
- The resulrs and discussion are also reasonable.
- Need a few minor changes. For example, authors can improve the description of the figures (e.g., title of y and x axis) and the tables.
Author Response
Response to Reviewer 2 Comments
Point 1: Need a few minor changes. For example, authors can improve the description of the figures (e.g., title of y and x axis) and the tables.
Response 1: Appropriate modifications have been made to the figures and tables, the titles of y and x axis have been added to the relevant pictures, and the meaning of asterisks in the tables has been described.